# Cytosine and adenine deaminase base-editors induce broad and nonspecific changes in gene expression and splicing

Jiao Fan[1,2,5], Yige Ding[3,5], Chao Ren[1], Ziguo Song[3], Jie Yuan[1], Qiuzhen Chen[1,4], Chenchen Du[3], Chao Li[3], Xiaolong Wang [3✉] & Wenjie Shu [1✉]

Cytosine or adenine base editors (CBEs or ABEs) hold great promise in therapeutic applications because they enable the precise conversion of targeted base changes without generating of double-strand breaks. However, both CBEs and ABEs induce substantial off-target DNA editing, and extensive off-target RNA single nucleotide variations in transfected cells. Therefore, the potential effects of deaminases induced by DNA base editors are of great importance for their clinical applicability. Here, the transcriptome-wide deaminase effects on gene expression and splicing is examined. Differentially expressed genes (DEGs) and differential alternative splicing (DAS) events, induced by base editors, are identified. Both CBEs and ABEs generated thousands of DEGs and hundreds of DAS events. For engineered CBEs or ABEs, base editor-induced variants had little effect on the elimination of DEGs and DAS events. Interestingly, more DEGs and DAS events are observed as a result of over expressions of cytosine and adenine deaminases. This study reveals a previously overlooked aspect of deaminase effects in transcriptome-wide gene expression and splicing, and underscores the need to fully characterize such effects of deaminase enzymes in base editor platforms.

[1] Department of Biotechnology, Beijing Institute of Radiation Medicine, Beijing, China. [2] Institute of Geriatrics, National Clinical Research Center of Geriatrics Disease, Second Medical Center of Chinese PLA General Hospital, Beijing, China. [3] Key Laboratory of Animal Genetics, Breeding and Reproduction of Shaanxi Province, College of Animal Science and Technology, Northwest A&F University, Yangling, China. [4] Computer School, University of South China, Hengyang, China. [5] These authors contributed equally: Jiao Fan, Yige Ding. ✉email: xiaolongwang@nwafu.edu.cn; shuwj@bmi.ac.cn

Recently, a variety of cytosine and adenine base editors (CBEs and ABEs) that combines cytosine or adenosine deaminases with CRISPR-Cas9, has been developed. This variety enables highly efficient and precise targeted C-to-T or A-to-G base conversions[1–4]. Previous studies have reported the induction of unexpected off-target DNA editing in mammalian cells by both CBEs and ABEs. This limited their broad use in biomedical studies and therapeutic applications[3,5]. Adeno-associated viruses (AAVs) are the most commonly utilized delivery system for gene therapies. The main reason is their efficiency of DNA editing, as well as their capability to maintain long-term gene expression without toxicity or immune responses to the viral vector in vivo[6–8]. Thus, the extent of the potential effects of DNA base editor-induced deaminases is of great importance. Recent studies have shown that both CBEs and ABEs can indeed induce extensive transcriptome-wide deamination of cytosine or adenosine in human cells; this leads to the generation of thousands of off-target RNA single nucleotide variations (SNVs)[9–13]. Whether the presence of this large number of unwanted RNA SNVs has genetic or transcriptional consequences has not been comprehensively explored to date, this lack of data may lead to underestimation of the effect. Although engineering-improved CBEs or ABEs can largely decrease the number of off-target RNA SNVs in human cells[11,12], a comprehensive characterization of the effects of CBE or ABE deaminases on RNA is crucial to fully realize their utility.

This study analyzed RNA-seq datasets from two recent studies[9,12]. Transcriptome-wide differentially expressed genes (DEGs) and differential alternative splicing (DAS) events that had been induced by CBEs or ABEs were identified. Both cytosine and adenine base editors had generated hundreds of DEGs and DAS events. However, these phenomena had previously not been reported. Furthermore, engineering of CBE or ABE variants had only negligible effects on the prevention of DEG and DAS occurrences and even increased the emergence of DEGs and DAS events because of the overexpression of cytosine and adenine deaminases. These data highlight a previously unreported aspect of deaminase effects on gene expression and splicing, which apparently cannot be eliminated by genetic engineering. These results have important implications for the future use of base editors in both research and as therapeutic agents, thus highlighting a full characterization of deaminase enzyme activities in base editors.

## Results

**Base editors induce transcriptome-wide DEGs**. To evaluate the deaminase effects of base editors, an RNA-seq based analysis of transcriptome-wide DEGs was conducted. This was applied to one type of CBE (BE3; APOBEC1-nCas9-UGI) and one type of ABE (ABE7.10; TadA-TadA*-nCas9), combined with GFP and either with or without a single guide RNA (sgRNA) in HEK293T cells. Cells receiving GFP alone were used as controls. We first compared GFP versus GFP cells and found the numbers of DEGs were closed to 0 (Supplementary Fig. 1a). A total of 1178, 757, 172, and 316 DEGs were identified in cells among APOBEC1, BE3 without sgRNA, BE3-site 3, and BE3-RNF2, respectively (Fig. 1a, b and Supplementary Data 1). Moreover, DEGs were found in both coding and non-coding regions (Supplementary Fig. 1b, c). Similarly, 1761, 859, 215, and 357 DEGs were identified in cells expressing only TadA-TadA*, ABE7.10 without sgRNA, and ABE7.10 with either site 1, or site 2 sgRNA, respectively (Fig. 1c, d, Supplementary Fig. 1d, e and Supplementary Data 1). To explore the relationship between off-target RNA SNVs and DEGs, we identified 17,826, 12,393, 6530, and

8893 putative RNA SNVs in CBEs and 7501, 3495, 3505, and 6661 putative RNA SNVs in ABEs, respectively (Supplementary and Methods Supplementary Tables 1 and 2). We noted that only about 5 and 10% of DEGs were mapped to RNA SNVs, respectively (Supplementary Tables 1 and 2). Notably, transfection of APOBEC1 or TadA-TadA* induced a higher number of DEGs compared with other transfected cells. This suggests that increased DEGs in CBE- or ABE-treated cells were likely caused by the overexpression of the deaminases APOBEC1 or TadA (Fig. 1a–d and Supplementary Data1). Moreover, the number of DEGs increased with higher levels of expressions of CBEs or ABEs (Fig. 1e, f). We observed 46.2 ± 9.0% (mean ± sem) or 52.4 ± 8.7% (mean ± sem) overlap between any two groups from the BE3- or ABE7.10-transfected cells with or without sgRNA, respectively (Fig. 1g, h). Together, the DEGs induced by CBEs and ABEs were independent of sgRNAs and were presumably a result of APOBEC1 and TadA-TadA* overexpression, respectively.

Next, we characterized these transcriptome-wide DEGs identified in BE3-treated cells. More than 60.0% of the DEGs were identified in BE3-treated or APOBEC1-expressing cells (Supplementary Fig. 1f). These data indicate that the occurrence of this substantial number of DEGs was, in fact, induced by BE3 or APOBEC1, rather than a spontaneous occurrence. Similarly, 59.3% of ABE7.10-induced DEGs were expressed, which is consistent with the action of TadA (Supplementary Fig. 1g). Notably, 12 and 20 cancer-associated genes were observed among the DEGs induced by BE3 and ABE7.10, respectively. These genes included *ETV5*, *CDKN1A*, and *DDIT3*, which highlights concerns about the oncogenic potential associated with DNA base editing (Fig. 1i, j). GO enrichment analysis of these DEGs identified in BE3- or ABE7.10-treated groups specifically identified enrichments of regulation of cell proliferation, angiogenesis, cell cycle arrest, and SMAD protein signal transduction (Supplementary Fig. 1h, i and Supplementary Data 2). Each of these pathways exerts well-established roles in the progression of cancer.

Bulk RNA-seq is based on large pools of cells, which offers the potential to mute random signals via population averaging. To account for this, single-cell RNA-seq analysis was performed of DEGs from each of the three groups[12] (GFP-alone, BE3-site 3, and ABE7.10-site 1; Fig. 1k). Hundreds of DEGs for BE3- or ABEs7.10-edited cells were consistently observed compared with the GFP-alone group (Fig. 1k). Notably, the percentage of DEGs shared by any of these BE3- or ABE7.10-edited cells (19.8 ± 1.6% or 15.5 ± 1.6%) was much lower than that of cell populations (52.4 ± 8.7%, 46.2 ± 9.0%) (Supplementary Fig. 1j). These data indicate that BE3- or ABE7.10-induced DEGs were essentially random and independent of the utilized sgRNA.

**Engineering of deaminases causes a modest reduction in DEGs**. Recent studies have reported that off-target RNA SNVs, induced by DNA base editors, can be eliminated via the engineering of deaminases[11,12]. To test whether such engineered deaminases of CBE or ABE variants can alleviate their off-target effects on gene expression, DEGs were identified in transfected HEK293T cells that expressed engineered CBE or ABE, variant groups. For CBE engineering, five CBE variants were examined: a point mutation and double mutations to BE3 (BE3^W90A and BE3^W90Y/R126E), BE3 with human APOBEC3A (hA3A), as well as its two variants BE3 (hA3A^R128A), and BE3 (hA3A^Y130F). These variants had a decreased number of off-target RNA SNVs while maintaining BE3-like DNA site-specific activity. BE3^W90A and BE3 (hA3A) induced a similar number of DEGs compared with BE3 (118, 123, and 172, respectively). BE3^W90Y/R126E, BE3(hA3A^R128A), and

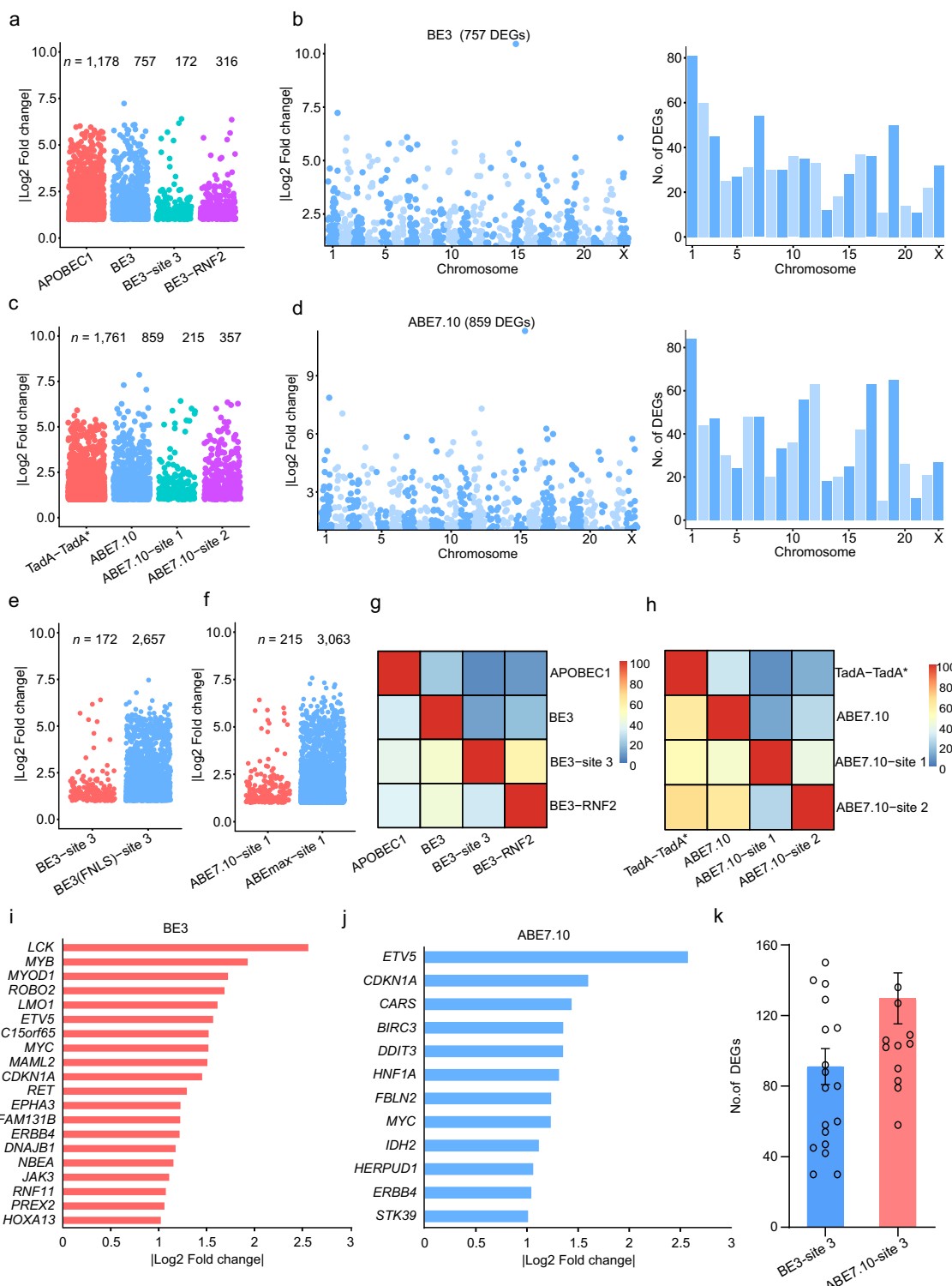

BE3(hA3A$^{Y130F}$) induced the largest numbers of DEGs compared with BE3 (1574, 3036, and 4524, respectively) (Fig. 2a, Supplementary Fig. 2a, b and Supplementary Data 3). More than 56.1% of the DEGs were identified in CBE variants-expressing cells (Supplementary Fig. 2c). For ABE engineering, the introduction of the mutations D53E or F148A into both TadA and TadA* showed that the numbers of DEGs in ABE7.10$^{D53E}$-transfected

cells increased relative to ABE7.10 with site 1 sgRNA (404 vs. 215 DEGs, respectively) (Fig. 2b, Supplementary Fig. 2d, e and Supplementary Data 3). The DEGs of ABE7.10$^{F148A}$-transfected cells at site 1 were slightly less than ABE7.10$^{D53E}$-transfected cells (132 vs. 215 DEGs, respectively). Surprisingly, ABE7.10$^{F148A}$- and site 2 sgRNA co-transfected cells induced 1854 DEGs (Fig. 2b). For ABE engineering, about 58.0% of DEGs occurred in the expressed

**Fig. 1 Identification and characterization of transcriptome-wide differentially expressed genes (DEGs) induced by DNA base editors. a** Jitter plots of RNA-seq experiments showing the DEGs identified in HEK293T cells transfected with APOBEC1, BE3, BE3-site 3, and BE3-RNF2. Cells transfected with a GFP-encoding plasmid served as a control for all comparisons. *n* total number of identified DEGs. **b** Manhattan plots showing representative distributions of DEGs across the human transcriptome in cells overexpressing BE3. Chromosomes are represented in different colors. The number of DEGs for each chromosome is presented at the right. **c** Jitter plots of RNA-seq experiments showing the DEGs identified in HEK293T cells transfected with TadA-TadA*, ABE7.10, ABE7.10-site 1, and ABE7.10-site 2. The GFP group serves as the control for all comparisons. *n* the total number of identified DEGs. **d** Manhattan plots showing the representative distributions of DEGs across the human transcriptome for ABE7.10. Chromosomes are indicated in different colors. The number of DEGs for each chromosome is presented at the right. **e** Jitter plots of RNA-seq experiments showing the DEGs identified in HEK293T cells transfected with BE3-site 3 or with BE3(FNLS)–site 3. GFP group served as the control for all comparisons. *n* total number of identified DEGs. **f** Jitter plots from RNA-seq experiments showing the DEGs identified in HEK293T cells transfected with ABE-site 1 or ABEmax–site 1. The GFP group served as a control for all comparisons. *n* total number of identified DEGs. **g, h** Ratio of shared DEGs between any two samples in the APOBEC1, BE3, TadA, and ABE7.10 overexpression groups. The proportion in each cell was calculated by the number of overlapping DEGs between two samples, divided by the number of DEGs in the row. **i, j** Cancer-related genes in BE3 or ABE7.10-induced DEGs. **k** Bar plots showing the number of identified DEGs in single-cell RNA-seq samples in BE3-site3 and ABE7.10-site1 overexpressing cells. Single-cell RNA-seq samples of GFP plasmid transfected cells served as a control for all comparisons. Error bars represent SD for three independent experiments. Corresponding data were presented in Supplementary Data 1.

genes (Supplementary Fig. 2f). Moreover, fewer ABEmax variants lacking the TadA domain (referred to as miniABEmax)[9] were generated, and two miniABEmax variants (K20A/R21A and V82G) were also generated. Each of these editors was sgRNA assayed targeting two endogenous sites (HEK site 2 and ABE site 16) and one site that does not occur in the human genome (non-targeting (NT)). DEGs of ABEmax, miniABEmax, miniABEmax (K20A/R21A), and miniABEmax (V82G) were identified. These four groups, at the three sgRNA targets, all induced thousands of DEGs when protein 2A (P2A)-enhanced GFP (EGFP) fusion (without nCas9) was used as control. Interestingly, ABEmax induced the highest number of DEGs compared with miniABEmax and the two miniABEmax variants (K20A/R21A and V82G) (Fig. 2c and Supplementary Data 4).

Next, the nCas9-dependence of ABE-mediated DEGs was examined. DEGs induced by ABEmax, miniABEmax, miniABEmax (K20A/R21A), and miniABEmax (V82G) were identified using three sgRNAs in conjunction with nCas9-UGI-NLS fusion. The four groups all induced hundreds of DEGs at particular sgRNA sites (Fig. 2d and Supplementary Data 5). With the exception of miniABEmax (V82G) at sgRNA site 2, ABEmax induced the most DEGs compared with miniABEmax and the two miniABEmax variants (K20A/R21A and V82G) (Fig. 2d). Moreover, a comparison of ABE variants with the GFP control showed that all four ABE variants yielded substantial reductions in DEGs compared with a nCsa9 with NT sgRNA negative control (Fig. 2c, d).

To examine the nCas9-dependence of BE3-mediated DEGs, the DEGs induced by three variants (BE3-R33A, BE3-R33A/K34A, and BE3-E63Q) were examined using *RNF2* gRNA. These three variants all had on-target DNA editing efficiencies comparable to that of wild-type BE3 with reduced RNA editing. The nCas9-UGI-NLS fusion was used as a negative control in this experiment. Modest decreases in the numbers of transcriptome-wide DEGs were observed. Expression of WT BE3 induced 1585 DEGs, and BE3-R33A or BE3-R33A/K34A induced 1145 and 1031 DEGs, respectively (Fig. 2e and Supplementary Data 6). Furthermore, we sought to determine whether CBEs harbor other cytidine deaminases (e.g., human APOBEC3A (hA3A), enhanced A3A (eA3A; an engineered A3A with more precise and specific DNA-editing activity), human activation-induced cytidine deaminase (hAID)) and whether these induce DEGs independent of nCas9. For this, DEGs in HEK293T cells transfected with plasmids expressing each of the CBEs (with sgRNA targeting the *RNF2* gene) were examined. By comparison, the two previously described variants R33A and R33A/K34A induced

slightly higher numbers of DEGs than eA3A-BE3 and hAID-BE3 (Fig. 2e and Supplementary Data 6) and more than 65% of DEGs were expressed, which may have confounding effects (Supplementary Fig. 2g).

Next, we examined whether the numbers of DEGs and off-target RNA SNVs are correlated. Overall, the numbers of off-target RNA SNVs accounted for approximately only 5.6% of the variation in the numbers of DEGs as identified by RNA-seq (Fig. 2f). The correlation in BE3-treated or ABE-treated groups was considered separately. The numbers of off-target RNA SNVs explained only 1.5% of the variation in CBE-treated groups, while it explained 31% in ABE-treated groups (Supplementary Fig. 2h, i). These data suggest that off-target RNA SNVs are likely not responsible for the significant observed transcriptional differences between groups treated with either base editors or controls.

**Base editors induce transcriptome-wide DAS events.** To test whether base editors cause extensive isoform dynamics, transcriptome-wide DAS events were identified and categorized into seven types: exon skipping (ES), alternative 5′ and 3′ splice sites (A5/A3), intron retention (RI), mutually exclusive exons (MX), and alternative first and last exons (AF/AL). We first identified DAS events between GFP groups by randomly selecting three repeats from six repeats of GFP cells as control groups and the other three repeats as treatment groups. The numbers of DAS events between GFP groups were mostly less than 100 (Supplementary Fig. 3a). In total, 1779 (APOBEC1), 568 (BE3 without sgRNA), 1821 (BE3-site 3), and 1612 (BE3-RNF2) DAS events were identified compared with GFP control groups (Fig. 3a and Supplementary Data 7). More than 90% of DAS events occurred in expressed genes (Supplementary Fig. 3b). Furthermore, 1949 (TadA-TadA*), 1324 (ABE7.10 without sgRNA), 513 (ABE7.10-site 1), and 1115 (ABE7.10-site 2) DAS events were identified (Fig. 3b and Supplementary Data 7) and also about 92.8% of DAS events were related to expressed genes (Supplementary Fig. 3c). Similarly, we observed that less than 5% of DAS events were mapped to RNA SNVs in CBE and ABE (Supplementary Tables 1 and 2). Next, the percentages of overlap in DAS events for CBE and ABE groups were examined. A total of 51.1 ± 8.1% (mean ± sem) and 56.8 ± 7.6% (mean ± sem) of overlaps between any two groups from the BE3- or ABE7.10-transfected cells were found, respectively, independent of the presence of sgRNA (Fig. 3c, d).

Moreover, thousands of DAS events were identified in RNA-seq samples from engineered CBE or ABE variant groups compared with GFP-transfected samples. For engineered CBEs,

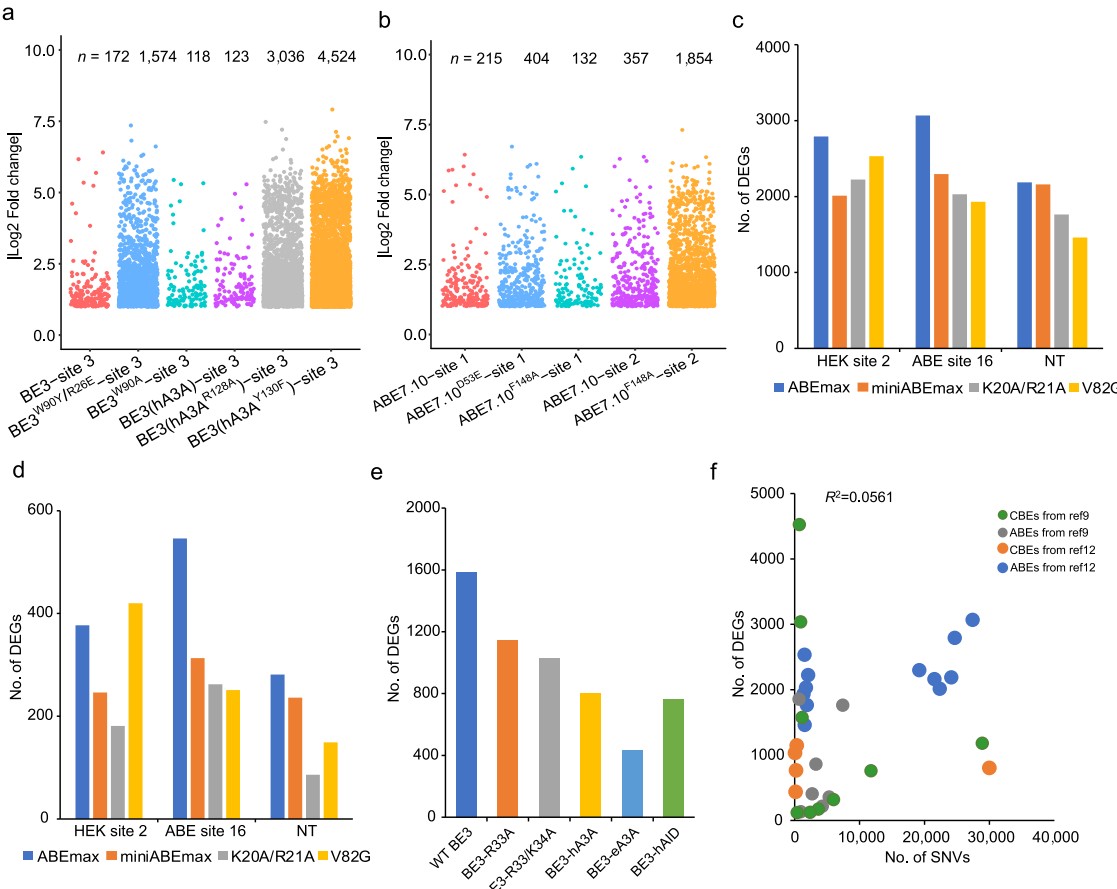

**Fig. 2 Identification of DEGs for different engineered base editor variants. a** Jitter plots of RNA-seq experiments in HEK293T cells showing DEGs identified for BE3-site 3, BE3$^{W90Y/R126E}$-site 3, BE3$^{W90A}$-site 3, BE3(hA3A)-site 3, BE3(hA3A$^{R128A}$)-site 3, and BE3(hA3A$^{Y130F}$)-site 3 groups. Cells transfected with GFP encoding plasmids serve as a control for all comparisons. *n* total number of identified DEGs. Corresponding data were presented in Supplementary Data 3. **b** Jitter plots from RNA-seq experiments in HEK293T cells showing the DEGs identified for ABE7.10-site 1, ABE7.10$^{D53E}$-site 1, ABE7.10$^{F148A}$-site 1, ABE7.10-site 2, and ABE7.10$^{F148A}$-site 2 groups. Cells transfected with GFP-encoding plasmids served as a control for all comparisons. *n* total number of identified DEGs. Corresponding data were presented in Supplementary Data 3. **c** Bar plots showing the number of DEGs identified in RNA-seq experiments in HEK293T cells that expressed ABEmax, miniABEmax, miniABEmax(K20A/R21A), or miniABEmax(V82G) as well as one of three different sgRNAs (HEK site 2, ABE site 16, and a non-targeted control (NT)). Cells transfected with GFP-encoding plasmids served as a control for all comparisons. Corresponding data were presented in Supplementary Data 4. **d** Bar plots showing the number of DEGs identified in RNA-seq experiments in HEK293T cells that expressed ABEmax, miniABEmax, miniABEmax(K20A/R21A), or miniABEmax(V82G) as well as one of three different gRNAs (HEK site 2, ABE site 16, and a non-targeted control (NT)). Cells expressing nCas9 with NT sgRNA served as a control for all comparisons. Corresponding data were presented in Supplementary Data 5. **e** Bar plots showing the number of identified DEGs in RNA-seq experiments in HEK293T cells that expressed WT BE3, SECURE-BE3(R33A), SECURE-BE3(R33A/K34A), hA3A-BE3, eA3A-BE3, and hAID-BE3 with co-expression of a gRNA, targeting a site on the RNF2 gene. Cells expressing nCas9 served as a control for all comparisons. Corresponding data were presented in Supplementary Data 6. **f** Correlations between the numbers of DEGs and the numbers of off-target RNA SNVs in BE3-treated groups. **g** Overall correlations between the numbers of DEGs and the numbers of off-target RNA SNVs in CBE-treated and ABE-treated groups. $R^2$ values were calculated using robust linear regressions on the numbers of DEGs and the numbers of off-target RNA SNVs.

DAS events of BE3$^{W90A}$ and BE3 (hA3A)-transfected cells at site 3 were less compared to BE3 (647, 1232, and 1821, respectively). In contrast, BE3 (hA3A$^{R128A}$) at the same three sites produced similar numbers of DAS events compared with BE3 (1785 vs. 1821) (Fig. 3e and Supplementary Data 8). For engineered ABEs, the numbers of DAS events in ABE7.10$^{D53E}$ site 1-transfected cells increased compared with that of ABE7.10-site 1 (1233 vs. 513, respectively) (Fig. 3f and Supplementary Data 8). When an F148A mutation was introduced into both TadA and TadA*, the DAS events of ABE7.10$^{F148A}$-transfected cells both at sites 1 and 2 increased compared with ABE7.10 (1334 vs. 513 or 1964 vs. 1115, respectively) (Fig. 3f and Supplementary Data 8). For these DAS events identified in engineered CBE or ABE variant groups,

the rates of expressed genes were also more than 90% (Supplementary Fig. 3d, e). Furthermore, a different study[11] reported that a large number of DAS events were induced by four ABE variants (ABEmax, miniABEmax, miniABEmax (K20A/R21A), and miniABEmax (V82G)) in HEK293 cells at site 2, ABE site 16, and non-targeting sgRNA compared with EGFP or nCsa9 controls (Fig. 3g, h, Supplementary Data 9 and Data 10). DAS events associated with BE3 variants (BE3-R33A, BE3-R33A/K34A, hA3A-BE3, eA3A-BE3, and hAID-BE3; all using the RNF2 gRNA) were observed, compared with the nCas9-UGI-NLS fusion negative control. These variants all decreased the number of DAS events compared with WT BE3 (Fig. 3i and Supplementary Data 11) and 93.6% of DAS events were related to expressed

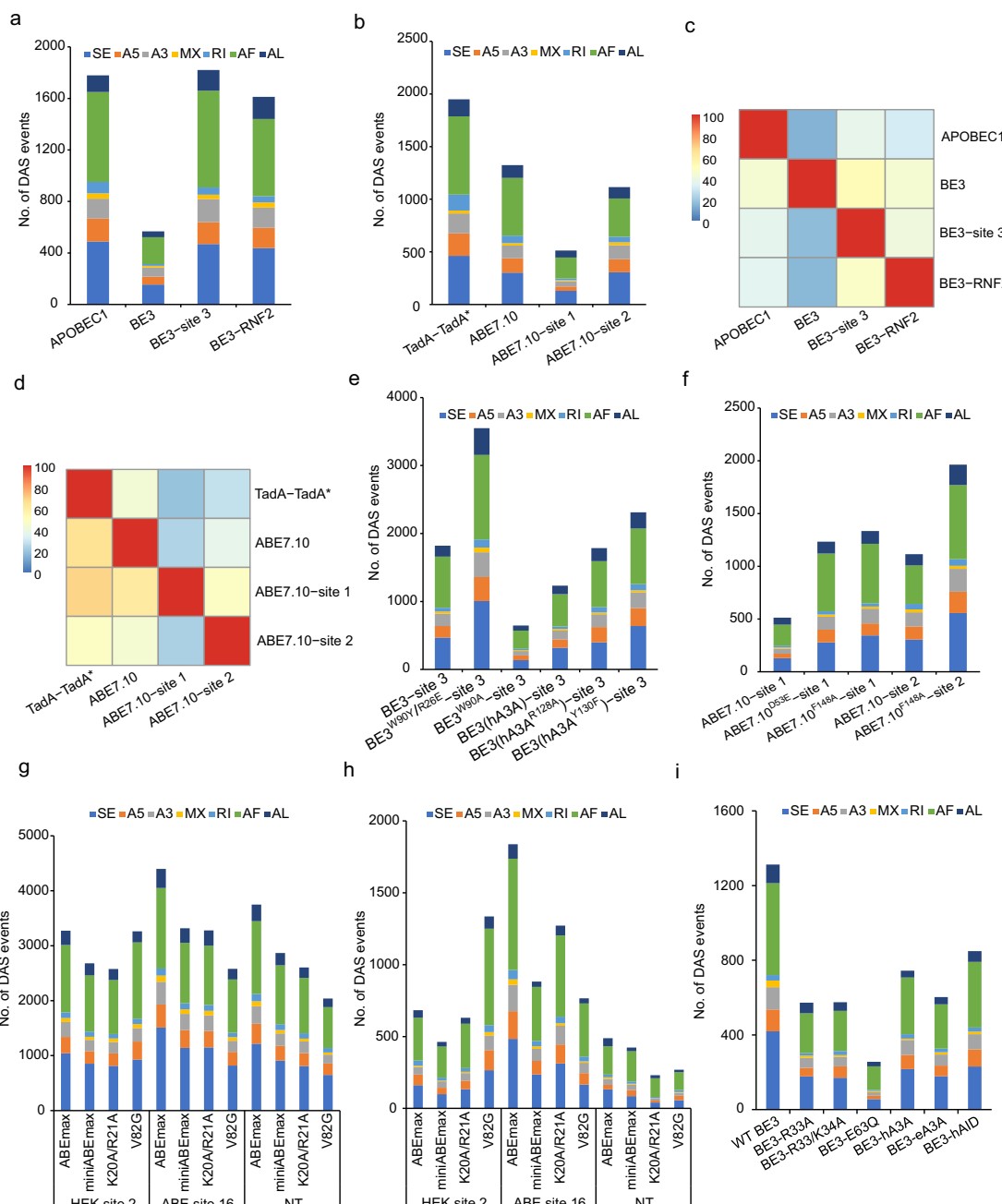

genes (Supplementary Fig. 3f). Thus, engineering CBE or ABE deaminases can only moderately decrease the frequency of DAS events.

**Validation of the identified DEGs and DAS in BE-treated cells**. To further validate DEGs and DAS identified in cells treated with base editors, we also separately performed RNA-seq in BE3 or TadA-TadA* transfected HEK293T cells. DEGs and DAS events were identified in comparison with GFP groups (Supplementary Fig. 4a). Totals of 1796 and 1151 DEGs were identified in BE3 and TadA-TadA* transfected HEK293T cells, including 41 and 36 cancer-related genes, respectively (Fig. 4a, b and Supplementary Fig. 4b, c). Totals of 1209 and 1518 DAS events were observed in BE3 and TadA-TadA*-treated groups, respectively (Fig. 4c). Several DEGs were randomly selected from each BE

editor and their expressional alterations were validated. These DEGs included *MYC*, *GPC3*, *SGK1*, *HSP90AA1*, *SETD2*, *PPM1D*, and *NCOA1*, which had been identified as oncogenes or tumor suppressor genes. For DEGs, differential expression was verified by qPCR (Fig. 4d, e and Supplementary Fig. 4d, e). For DAS, several potential candidates of skipped exons were identified: *RBM15*, *BAG4*, *ADRM1*, *TCEAL9*, and *PPP5C* for BE3-treated groups as well as *UBE2I*, *ISCU*, *RIF1*, *GADD45A*, and *APEH* for TadA-TadA*-treated groups. PCR analyses of the GFP groups and base editor-treated groups were performed using primers flanking the skipped exons of these target genes. The results verified that the exon inclusion levels of these genes were all significantly changed in response to treatment with the base editor (Fig. 4f–i, Supplementary Fig. 5a–f, and Supplementary Fig. 7a). In summary, these results suggest that large numbers of

**Fig. 3 Transcriptome-wide differential alternative splicing (DAS) events induced by DNA base editors. a** Bar plots from RNA-seq experiments showing identified DAS events in HEK293T cells expressing APOBEC1, BE3, BE3-site 3, and BE3-RNF2. Cells expressing GFP served as a control for all comparisons. Corresponding data were presented in Supplementary Data 7. **b** Bar plots from RNA-seq experiments showing identified DAS events in HEK293T cells transfected with TadA-TadA*, ABE7.10, ABE7.10-site 1, and ABE7.10-site 2. Cells expressing GFP served as a control for all comparisons. Corresponding data were presented in Supplementary Data 7. **c**, **d** Ratio of shared DAS events between any two samples in APOBEC1, BE3, TadA, and ABE7.10 groups. The proportion in each cell was calculated by the number of overlapping DAS events between two samples divided by the number of DAS events in the row. **e** Bar plots of RNA-seq experiments in HEK293T cells showing the identified DAS events for BE3-site 3, BE3$^{W90Y/R126E}$-site 3, BE3$^{W90A}$-site 3, BE3 (hA3A)-site 3, BE3(hA3A$^{R128A}$)-site 3, and BE3(hA3A$^{Y130F}$)-site 3 groups. GFP expressing cells served as a control for all comparisons. Corresponding data were presented in Supplementary Data 8. **f** Bar plots from RNA-seq experiments in HEK293T cells showing the identified DAS events for ABE7.10-site 1, ABE7.10$^{D53E}$-site 1, ABE7.10$^{F148A}$-site 1, ABE7.10-site 2, and ABE7.10$^{F148A}$-site 2 groups. GFP expressing cells served as a control for all comparisons. Corresponding data were presented in Supplementary Data 8. **g** Bar plots showing the number of identified DAS events in RNA-seq experiments in HEK293T cells expressing ABEmax, miniABEmax, miniABEmax(K20A/R21A), or miniABEmax(V82G) as well as one of three different gRNAs (HEK site 2, ABE site 16, and a non-targeted control (NT)). Cells expressing GFP served as a control for all comparisons. Corresponding data were presented in Supplementary Data 9. **h** Bar plots showing the number of identified DAS events in RNA-seq experiments in HEK293T cells expressing ABEmax, miniABEmax, miniABEmax(K20A/R21A), or miniABEmax(V82G) as well as one of three different sgRNAs (HEK site 2, ABE site 16, and a non-targeted control (NT)). Cells transfected with nCas9 and NT sgRNA served as a control for all comparisons. Corresponding data were presented in Supplementary Data 10. **i** Bar plots showing the number of identified DAS events in RNA-seq experiments in HEK293T cells expressing WT BE3, SECURE-BE3(R33A), SECURE-BE3(R33A/K34A), hA3A-BE3, eA3A-BE3, and hAID-BE3 with co-expression of a sgRNA, targeting a site on the RNF2 gene. nCas9 expressing cells served as a control for all comparisons. Corresponding data were presented in Supplementary Data 11.

DEGs and DAS events exist universally across base editor-treated cells.

**Base editors induce transcriptome-wide DEGs and DAS events in other mammalian cell types.** We also tested the applicability of our findings to other mammalian cell types. We investigated the changes in gene expression and splicing induced by base editors in HeLa cells (Supplementary Fig. 4a). There were 1386 and 732 DEGs identified in BE3 or TadA-TadA* transfected HeLa cells, including 34 and 31 cancer-related genes, respectively (Fig. 5a, b, Supplementary Fig. 6a, b, and Supplementary Data 13). Also, a total of 2035 and 1263 DAS events were observed in BE3 and TadA-TadA*-treated HeLa cells, respectively (Fig. 5c and Supplementary Data 3). We compared the DEG and DAS events between HEK293T and HeLa cells and found that only a few of DEG and DAS events in these two cells overlapped with each other, respectively (Supplementary Fig. 6c). It was well consistent with the observations of off-targets SNVs between different cell types. In the same way, we randomly selected several DEGs, including *ETV5*, *HSPE1*, *DUSP1*, *DNAJB1*, *SERPINH1*, and *DDIT3*, and verified the differential expression in HeLa cells by qPCR (Fig. 5d, e). For DAS, several potential candidates of skipped exons were identified by PCR analyses using primers flanking the skipped exons of these target genes, including *QPRT* and *ZNHIT1* for BE3-treated HeLa cells as well as *DHX9* and *RLIM* for TadA-TadA*-treated cells (Fig. 5f–i and Supplementary Fig. 7b). Together, these data indicate that the transcriptome-wide DEGs and DAS events induced by base editors ubiquitously occur in mammalian cells.

**Increased expression of deaminases induces an increase in DEGs and DAS events.** Because our data have suggested that higher expressing level of CBEs or ABEs trend to induce more DEGs or DAS events, we hypothesized the potential effect of the expression levels of deaminases in gene expression and alternative splicing. Note that this hypothesis does not preclude the possibility that there are other potential mechanisms for these widespread changes induced by base editors. To test this hypothesis, HEK293T cells were transfected with a gradient concentration of APOBEC1 (0 µg, 1.5 µg, 2.5 µg, and 3.5 µg), and the expression of APOBEC1 in cells was confirmed by qPCR (Fig. 6a and Supplementary Fig. 4a). Then RNA-seq has performed in these gradient concentration-transfected cells. We also calculated

FPKM values of APOBEC1 for RNA-seq data and observed the gradient expression of deaminases in HEK293T cells (Fig. 6b). We found that the number of DEGs was increased when higher levels of APOBEC1 were expressed. There were 0, 990, 2119, and 2581 DEGs identified in HEK293T cells transfected with 0 µg, 1.5 µg, 2.5 µg, or 3.5 µg APOBEC1, respectively (Fig. 6c and Supplementary Data 14). Totals of 146, 652, 1061, and 1346 DAS events were observed in gradient concentration of APOBEC1-transfected groups, respectively (Fig. 6c, Supplementary Data 14). Simultaneously, we examined the potential effect of expression of ABE deaminases by performing gradient concentration transfection of TadA-TadA* in HEK293T cells (Supplementary Fig. 4a) and confirmed the expression levels of deaminases in each group (Fig. 6d, e). We also observed an increase in numbers in DEGs and DAS events induced by increased expression of ABE deaminases (Fig. 6f, Supplementary Data 14). There were 0, 1264, 1076, and 2312 DEGs and 146, 956, 1384, and 1607 DAS events identified in HEK293T cells transfected with 0 µg, 1.5 µg, 2.5 µg, or 3.5 µg TadA-TadA*, respectively (Fig. 6f). Taken together, our results confirmed our supposition that a higher expressing level of CBEs or ABEs will result in more DEGs and DAS events.

**Discussion**

Recent developments of DNA base editors (both CBEs and ABEs) resulted in the rapid emergence of state-of-the-art tools for genome editing in model organisms. Despite the advantages, these base-editing methods offer, several potential issues, such as target specificity, off-target DNA edits, and the recently reported base editor-induced off-target RNA SNVs, impede the use of these rapidly evolving techniques for biomedical applications. To alleviate these disadvantages, further efforts to more fully characterize the potential effects of deaminases are critical. This study comprehensively examined the deaminase effects of CBEs or ABEss on RNA. To this end, transcriptome-wide DEGs and DAS events induced by CBEs or ABEs were examined in RNA-seq samples[9,12]. The data presented here show that canonical CBEs and ABEs generated substantial numbers of DEGs and DAS events, which were independent of sgRNA, and appear to be caused by overexpression of APOBEC1 and TadA-TadA*, respectively.

Notably, recent studies have shown that the engineering of deaminases (both CBE and ABE variants) may eliminate incidences of off-target SNVs without substantially affecting on-

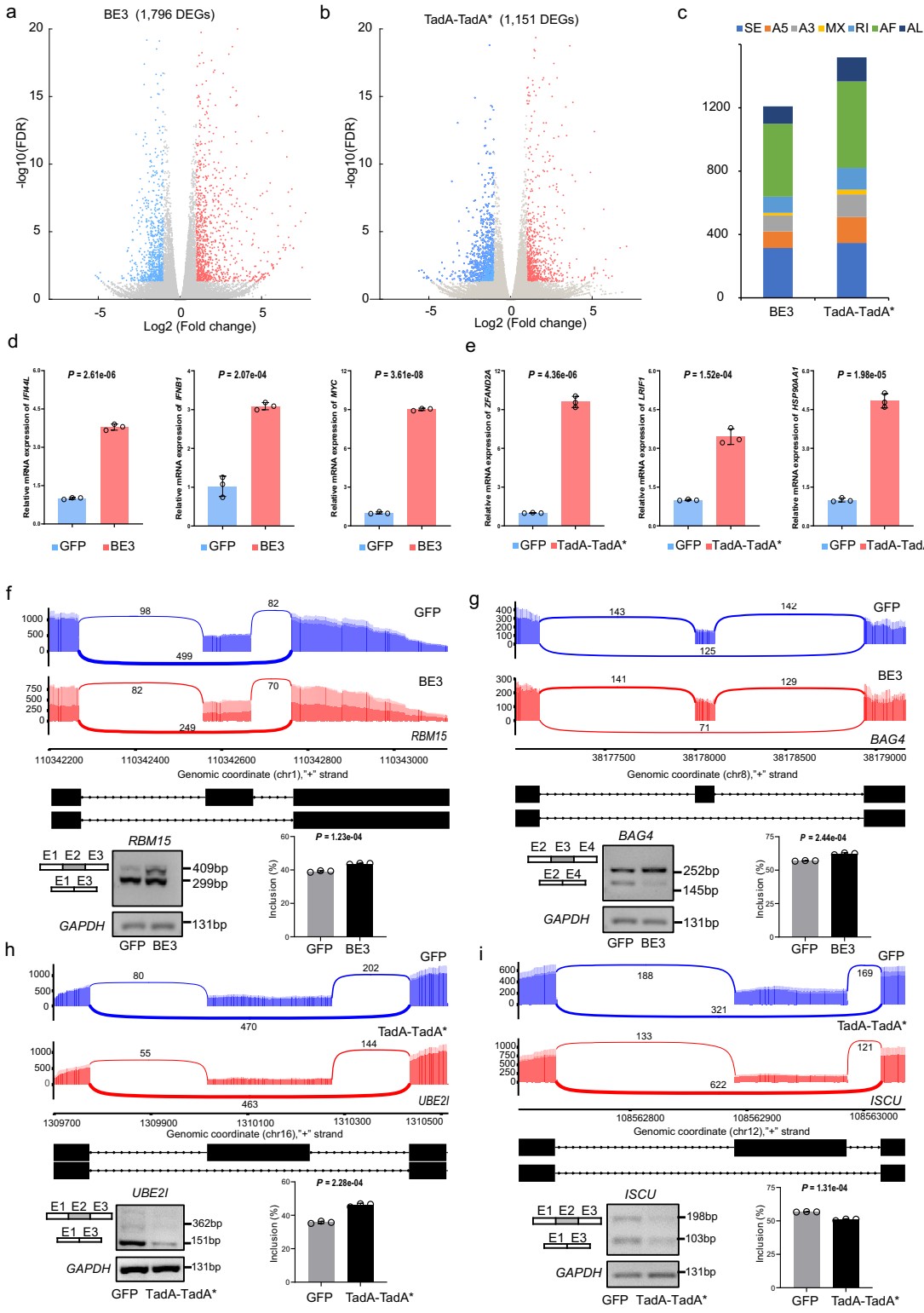

target DNA editing[12]. However, in the present study, both CBE and ABE variants failed to adequately decrease transcriptome-wide DEGs and DAS events. In fact, hundreds to thousands of DEGs and DAS events were generated, respectively. Although the presence of a few high-fidelity variants (i.e., BE3(hA3AR128A), BE3(hA3AY130F), and ABE7.10[F148A]-site 2) showed significantly fewer off-target RNA SNVs than canonical BEs in vitro, they induced significantly more DEGs and DAS events. In addition, these DEGs and DAS events were independent of Cas9, suggesting that they were the result of deaminase overexpression alone.

Genetic correction of inherited diseases depends largely on a single delivery system, which is most commonly used: AAVs. These viruses exhibit prolonged gene expression in vivo[6–8]. Thus, the fact that base editors (i.e., CBEs or ABEs) and their variants continuously generate extensive DEGs and DAS events overtime

**Fig. 4 Validation of DEGs and DAS events identified from RNA-seq data in HEK293T cells. a** Volcano plots showing the significant DEGs (adjusted *p*-value ≤ 0.05 and |Fold change| ≥ 2; downregulated (blue), upregulated (red), and unchanged genes (gray)) in BE3-treated groups. Cells expressing GFP served as control. **b** Volcano plots showing the significant DEGs (adjusted *p*-value ≤ 0.05 and |Fold change| ≥ 2; downregulated (blue), upregulated (red), and unchanged genes (gray)) in TadA-TadA*-treated groups. Cells expressing GFP served as control. **c** Bar plots from RNA-seq experiments showing identified DAS events in HEK293T cells transfected with BE3 and TadA-TadA*. Cells expressing GFP served as a control for all comparisons. Corresponding data of (**a**–**c**) were presented in Supplementary Data 12. **d** DEG validation in BE3-treated groups by qPCR. The tested genes were randomly selected and the RNA levels are shown relative to those of the control GFP group, normalized to 1. Data of qPCR are presented as mean ± sem, *n* = 3 biologically independent samples. **e** DEG validation in TadA-TadA*-treated groups by qPCR. The tested genes were randomly selected and the RNA levels are shown relative to those of the control GFP group, normalized to 1. Data of qPCR are presented as mean ± sem, *n* = 3 biologically independent samples. **f** Sashimi plot and PCR validation of the RBM15 gene in GFP and BE3-treated cells. The alternative exon (Exon 2 (E2)) is marked in gray. **g** Sashimi plot and PCR validation of the BAG4 gene in GFP and BE3-treated cells. The alternative exon (Exon 3 (E3)) is marked in gray. **h** Sashimi plot and PCR validation of the UBE2I gene in GFP and TadA-TadA*-treated cells. The alternative exon (Exon 2 (E2)) is marked in gray. **i** Sashimi plot and PCR validation of the ISCU gene in GFP and TadA-TadA*-treated cells. The alternative exon (Exon 2 (E2)) is marked in gray. Error bars represent SD for three independent experiments.

to come poses additional risks in gene therapies, and oncogenesis is of particular concern in this context. Both BE3- and ABE7.10-treated groups showed a dramatic increase in the levels of *ETV5* (Ets variant gene 5; also known as ERM). This variant has been associated with metastatic progression in several types of human cancers[14,15]. Differential expression of the transcriptional target of the tumor suppressor *TP53* (encoding p53) cyclin-dependent kinase inhibitor 1A (*CDKN1A*; encoding p21) was found[16]. To further validate DEGs and DAS identified in base editor-treated cells, RNA-seq was conducted and the expression of genes and AS events in base editor-treated cells were detected. The significantly up-regulated expression of *MYC* was confirmed to be induced by BE3. The *MYC* oncogene is overexpressed in more than half of human cancers[17]. The confounding effects of unwanted DEGs and DAS events must be accounted for in relevant studies, especially if constitutive base editor expression is applied. For therapeutic applications in humans, both the duration and level of base editor deaminase expression should be minimized.

Recently, the first 3-D cryo-electron microscopy structure of base editors has been reported, which enables a better understanding of the molecular basis for DNA adenosine deamination by ABEs[18]. Activity assays indicated why ABEs are prone to create off-target edits: in ABEs, the deaminase protein fused to Cas9 is always active. As Cas9 hops around the nucleus, it binds and releases hundreds or even thousands of DNA segments before it finds its intended target. The attached deaminase does not wait for a perfect match and often edits a base before Cas9 finds its final target. Maybe the high activity of fused deaminase proteins can partly explain the transcriptome-wide effects on gene expression and alternative splicing. In fact, we indeed found that increased expression of deaminases induces an increased number in DEGs and DAS events. Future studies are necessary to explore the proposed mechanism and identify potential unanticipated activities of base editors.

The rapid evolution of genome editing technologies has led us into a new era. We can now easily edit our genomes and as well as the genomes of many other organisms. Expect for off-target RNA SNVs effect induced by genome-editing technologies reported by recent studies, we show here a critical and previously unreported aspect of the effects deaminases impose on gene expression and alternative splicing. We present further insight into the current understanding of the potential effects of deaminases induced by DNA base editors, which has important implications for the application of these technologies in both research and clinical settings. The most basic requirement of application the of genome-editing technologies in clinical treatment is accuracy and safety. Safety assessments for human therapeutic applications should include a comprehensive evaluation of the potential functional consequences induced by transcriptome-wide DEGs

and DAS events. Although these DEGs or DAS events could exist for only a short period of time by transient expression of base editors, the longer-term functional consequences of widespread unwanted gene expression and alternative splicing will need to be accounted for in research studies.

## Methods

**RNA-seq data collection, quality control, and processing**. The raw sequencing data of cells treated with CBE and ABE, in addition to those treated with engineered base editor variants were downloaded from the Sequence Read Archive BioProject (accession number PRJNA528149) and Gene Expression Omnibus (GEO, accession number GSE129894). Single-cell RNA-seq data for CBE and ABE groups were downloaded from SRA (accession number PRJNA528561). The quality of RNA-seq data was controlled using FastQC (v0.11.8). The RNA-seq data was trimmed with Trim Galore (v0.6.1) using default parameters for pair-end data to remove both low-quality reads and adaptors. The trimmed RNA-seq data were aligned to the GRCh38 human transcriptome with STAR (v2.5.0a)[19] using default parameters. Gene expression was quantified as fragments per kilobase of transcript per million mapped reads (FPKM) in HTSeq (v0.10.0)[20].

**Transient transfection and sequencing**. Plasmids were constructed with ClonExpress II One Step Cloning Kit (Vazyme) following the standard protocol. HEK293T (ATCC CRL-3216) and HeLa (ATCC CCL-2) were obtained from the American type culture collection (ATCC). Mycoplasma contamination was determined by PCR of the supernatant of HEK293T or HeLa cells. HEK293T cells and HeLa cells were seeded in 10-cm dishes and cultured in Dulbecco's modified Eagle's medium (DMEM, Thermo Fisher Scientific) supplemented with 10% FBS (Bi) and penicillin–streptomycin at 37 °C with 5% $CO_2$. Cells were transfected with 30 μg plasmids using Lipofectamine 3000 (Thermo Fisher Scientific). Three days after transfection, cells were digested with 0.05% trypsin and were prepared for FACS. GFP-positive cells were sorted and retained in Trizol (Solarbio). For RNA-seq, about 1,000,000 cells (the top 5% according to GFP signal strength) were collected and their RNA was extracted following the standard protocol. To construct the library, mRNAs were fragmented and converted into cDNA using random hexamers or oligo(dT) primers. The 5′ and 3′ ends of cDNA were ligated with adaptors, and correctly ligated cDNA fragments were enriched and amplified by PCR. For HEK293T cells, nine groups of transfected cells were used: cells that expressed only GFP, TadA-TadA*, or BE3. Data for GFP and BE3 are shown in Fig. 4a, c, Supplementary Data 12, and data for GFP and TadA-TadA* are shown in Fig. 4b, c, Supplementary Data 12. For HeLa cells, eight groups of transfected cells were used: cells that expressed only GFP, TadA-TadA*, or BE3. Data for GFP and BE3 are shown in Fig. 5a, c, Supplement Data 13, and data for GFP and TadA-TadA* are shown in Fig. 5b, c, Supplementary Data 13. For gradient concentration transfection assays, 0 μg, 1.5 μg, 2.5 μg and 3.5 μg APOBEC1 or TadA-TadA* was transfected in HEK293T cells, and data for HEK293T cells transfected with different concentration of APOBEC1 are shown in Fig. 6a–c, Supplementary 14 and data for HEK293T cells transfected with different concentration of TadA-TadA* are shown in Fig. 6d–f, Supplementary 14. High-throughput mRNA sequencing was conducted using Nova-PE150. The process of RNA-seq data was performed using the same analysis pipeline mentioned above. All sequencing data have been deposited under GEO.

**Identification of DEGs in base editor-treated cells**. DEGs were identified using DESeq2 (V1.20.0)[21]. A gene was identified as a DEG if changes in expression were > 2 or <0.5. The statistical significance of differential expression was evaluated using a Student's *t*-test on-base editor-treated vs. GFP or nCas9 groups. The results were filtered according to a false discovery rate of <0.05 using the Benjamini–Hochberg procedure.

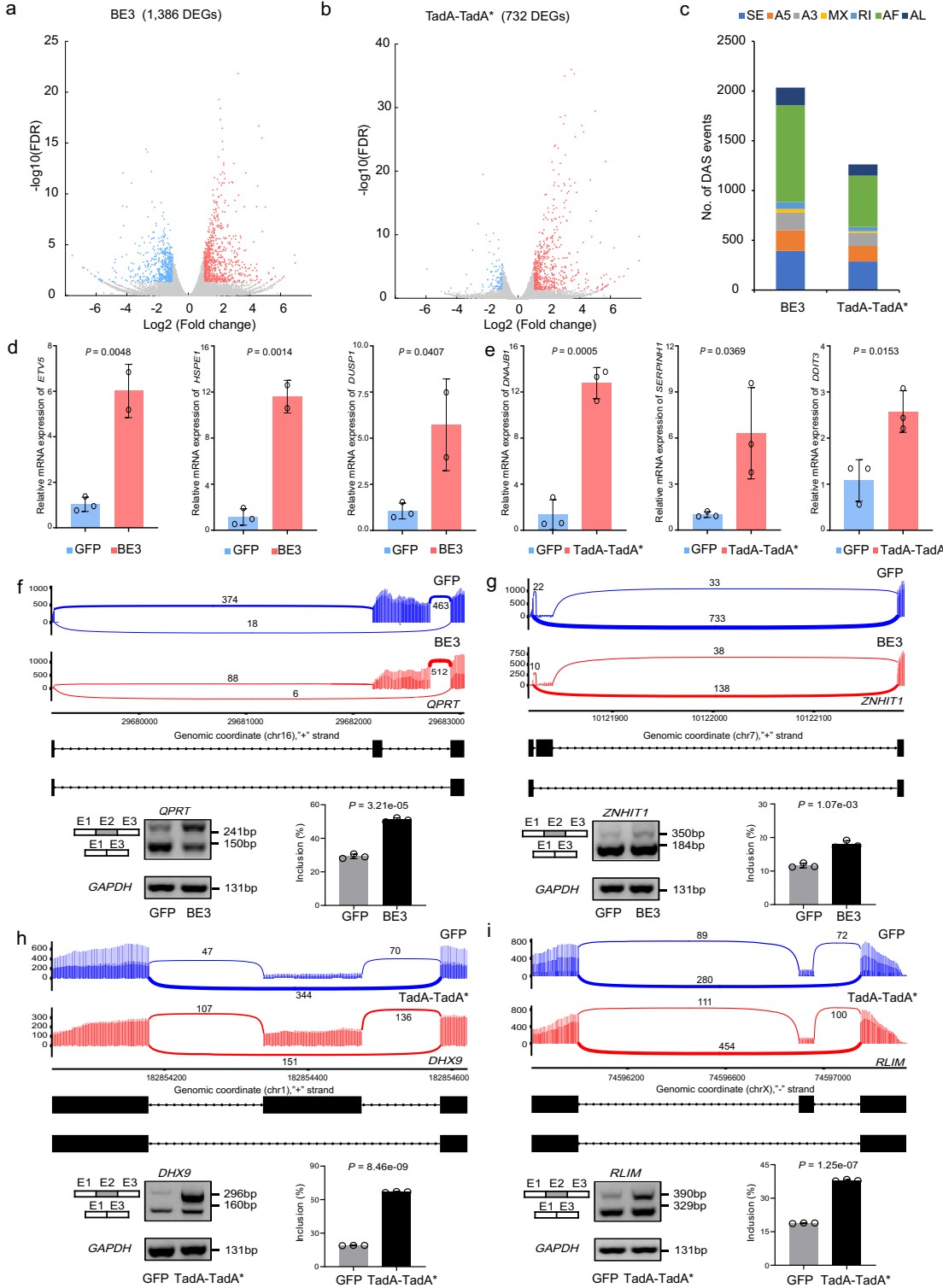

## Identification of DAS events in base editor-treated cells.
For DAS analysis, clean data were mapped to the GRCh38 reference genome using default parameters in SUPPA2 (v2.2.1)[22]. Seven types of DAS events (SE, A5/A3, RI, MX, and AF/AL) were identified, using a false discovery rate <0.05.

## Validation of DEGs and DAS events in base editor-treated cells.
Total RNA was extracted from transfected cells using the TRIzol reagent (Invitrogen). First-strand cDNA synthesis was performed using the revertaid first strand cDNA synthesis kit (Thermo Fisher Scientific). For qPCR, the TransStart® Green qPCR SuperMix (Trans) was used following the manufacturer's instructions. Amplification was performed in a LightCycle®96 real-time PCR system (Roche).

Quantification of fold changes in gene expression was calculated using the comparative threshold cycle (Ct) method with *GAPDH* as internal control based on the following equation: fold change (test vs. control) = $2^{-\Delta\Delta Ct}$, where $\Delta\Delta Ct = (Ct^{test/gene} - Ct^{control/gene}) - (Ct^{test/GAPDH} - Ct^{control/GAPDH})$. The sequences of all qPCR primers used in this study are listed in Supplementary Table 1. Validation of DAS events was performed by PCR analysis using primers (listed in Supplementary Data 15) flanking the skipped exons of target genes.

## Statistics and reproducibility.
All values are presented as mean ± sem. Unpaired Student's *t*-tests (two-tailed) were used for comparisons, and *P* < 0.05 was

**Fig. 5 Validation of DEGs and DAS events identified from RNA-seq data in HeLa cells. a** Volcano plots showing the significant DEGs (adjusted *p*-value ≤ 0.05 and |Fold change| ≥ 2; downregulated (blue), upregulated (red), and unchanged genes (gray)) in BE3-treated groups. Cells expressing GFP served as control. **b** Volcano plots showing the significant DEGs (adjusted *p*-value ≤ 0.05 and |Fold change| ≥ 2; downregulated (blue), upregulated (red), and unchanged genes (gray)) in TadA-TadA*-treated groups. Cells expressing GFP served as control. Corresponding data were presented in Supplementary Data 13. **c** Bar plots from RNA-seq experiments showing identified DAS events in HeLa cells transfected with BE3 and TadA-TadA*. Cells expressing GFP served as a control for all comparisons. Corresponding data were presented in Supplementary Data 3. **d** DEG validation in BE3-treated groups by qPCR. The tested genes were randomly selected and the RNA levels are shown relative to those of the control GFP group, normalized to 1. Data of qPCR are presented as mean ± sem, *n* = 3 biologically independent samples. **e** DEG validation in TadA-TadA*-treated groups by qPCR. The tested genes were randomly selected and the RNA levels are shown relative to those of the control GFP group, normalized to 1. Data of qPCR are presented as mean ± sem, *n* = 3 biologically independent samples. **f** Sashimi plot and PCR validation of the QPRT gene in GFP and BE3-treated cells. The alternative exon (Exon 2 (E2)) is marked in gray. **g** Sashimi plot and PCR validation of the ZNHIT1 gene in GFP and BE3-treated cells. The alternative exon (Exon 2 (E2)) is marked in gray. **h** Sashimi plot and PCR validation of the DHX9 gene in GFP and TadA-TadA*-treated cells. The alternative exon (Exon 2 (E2)) is marked in gray. **i** Sashimi plot and PCR validation of the RLIM gene in GFP and TadA-TadA*-treated cells. The alternative exon (Exon 2 (E2)) is marked in gray. Error bars represent SD for three independent experiments.

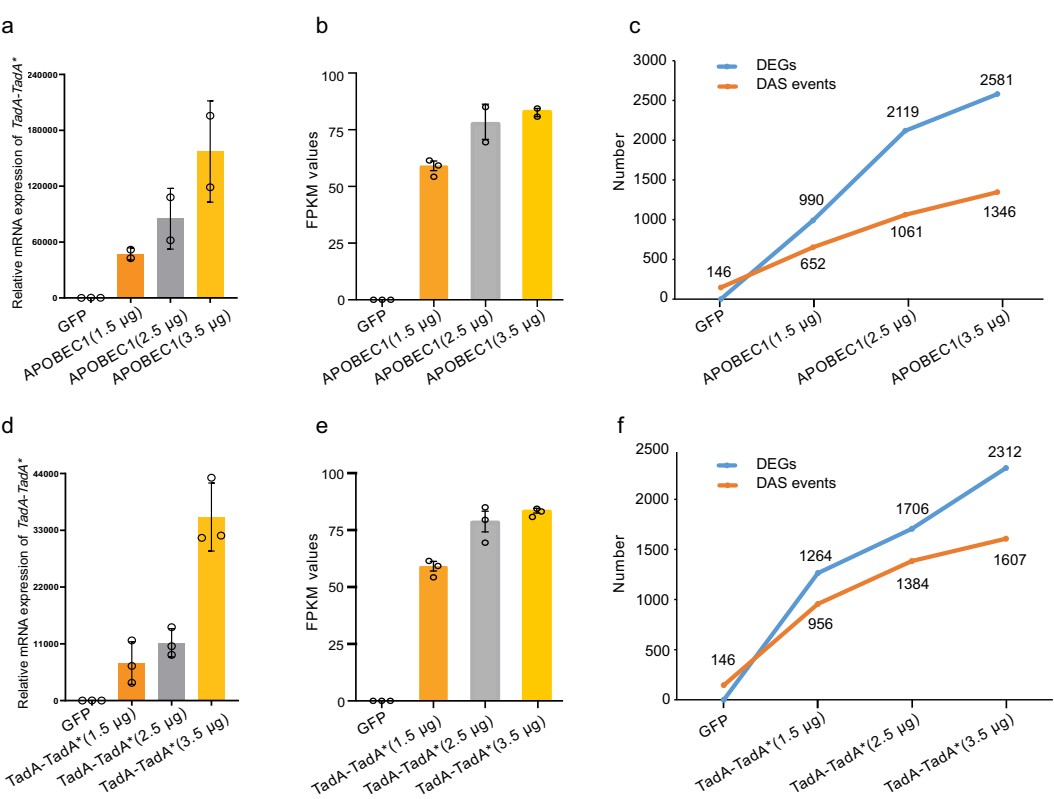

**Fig. 6 Increased expression of deaminases induces an increase in DEGs and DAS events. a** Expression of APOBEC1 detected by qPCR in HEK293T cells transfected with a gradient concentration of APOBEC1 (0, 1.5, 2.5, and 3.5 μg). All values are presented as mean ± sem. **b** FPKM values of APOBEC1 calculated by RNA-seq data in HEK293T cells transfected with a gradient concentration of APOBEC1 (0, 1.5, 2.5, and 3.5 μg). All values are presented as mean ± sem. **c** The number of DEGs and DAS events in cells transfected with a gradient concentration of APOBEC1 (0, 1.5, 2.5, and 3.5 μg). Corresponding data were presented in Supplementary Data 14. **d** Expression of TadA detected by qPCR in HEK293T cells transfected with a gradient concentration of TadA-TadA* (0, 1.5, 2.5, and 3.5 μg). All values are presented as mean ± sem. **e** FPKM values of TadA calculated by RNA-seq data in HEK293T cells transfected with a gradient concentration of TadA-TadA* (0, 1.5, 2.5, and 3.5 μg). All values are presented as mean ± sem. **f** The number of DEGs and DAS events in cells transfected with a gradient concentration of TadA-TadA* (0, 1.5, 2.5, and 3.5 μg). Error bars represent SD for two or three independent experiments.

considered to indicate statistically significant differences. A number of repeated experiments was described in the figure legends.

**Reporting summary**. Further information on research design is available in the Nature Research Reporting Summary linked to this article.

## Data availability

The authors declare that all relevant data supporting the findings presented in this study are available within the article and its Supplementary Information files, or, from the corresponding author upon reasonable request. The raw sequencing data of cells treated with CBE and ABE, in addition to those treated with engineered base editor variants were downloaded from the Sequence Read Archive BioProject (accession number PRJNA528149) and Gene Expression Omnibus (GEO, accession number GSE129894). Single-cell RNA-seq data for CBE and ABE groups were downloaded from SRA (accession number PRJNA528561). Fig. 1 and Supplementary Fig. 1a–e correspond to Supplementary Data 1. Supplementary Fig. 1h, i correspond to Supplementary Data 2. Fig. 2a, b, Supplementary Fig. 2a, b, d, and Fig. 5c correspond to Supplementary Data 3. Fig. 2c corresponds to Supplementary Data 4. Fig. 2d corresponds to Supplementary Data 5. Fig. 2e corresponds to Supplementary Data 6. Fig. 3a, b correspond to Supplementary Data 7. Fig. 3e, f correspond to Supplementary Data 8. Fig. 3g corresponds to Supplementary Data 9. Fig. 3h corresponds to Supplementary Data 10.

Fig. 3i corresponds to Supplementary Data 11. Fig. 4a–c correspond to Supplementary Data 12. Fig. 5a–c and Supplementary 6a, b correspond to Supplementary Data 13. Fig. 6c corresponds to Supplementary Data 14.

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

## Acknowledgements

This work was jointly supported by the National Natural Science Foundation of China (61873276, 31972526, 31772571). We thank Professor Hui Yang at Shanghai Institutes for Biological Sciences, Chinese Academy of Sciences, and Professor Branden S. Moriarity at Masonic Cancer Center and Center for Genome Engineering, the University of Minnesota for technical and analytical support.

## Author contributions

W.S. and X.W. designed research; J.F., Y.D., C.R., Z.S., J.Y., Q.C., C.D., and C.L. performed research; W.S., J.F., and X.W. wrote the manuscript. All authors read and approved the manuscript.

## Competing interests

The authors declare no competing interests.
