## [Peer Review File · Communications Biology]

Reviewers' Comments:

Reviewer #1:

Remarks to the Author:

Previous studies showed that cytosine or adenine base editors (CBEs or ABEs) induced substantial off-target effects at both DNA and RNA levels, which raised serious concerns about the therapeutic applications of these genome editing tools. In this study, the authors examined the transcriptome-wide deaminase effects on gene expression and found that both CBEs and ABEs generated hundreds to thousands differentially expressed genes (DEGs) and differential alternative splicing (DAS) events. They further examined the previously reported high-fidelity base editors and found that these editors only showed limited effects on the reduction of DEGs and DAS. Overall, I think this study reveals a very important aspect of base editor-induced off-target effects. However, the authors missed an essential control group for almost all sets of experiments: how many DEGs and DAS events could be induced by comparing GFP v.s. GFP cells (for reference, see PMID: 31181567)? This control group must be included for all comparisons.

Reviewer #2:

Remarks to the Author:

In this study, the authors systematically analyzed the potential impact of ABEs and CBEs on gene expression and alternative splicing. They found that both editors could induce hundreds of DEGs and DAS events, which seems independent of cas9 expression. This is an interesting topic to look into; however, the current results are too preliminary to draw any conclusions.

Major points:

The fold changes of DEGs and DAS events are only informative when used in conjunction with their gene expression levels. For example, if most of these events occur in lowly expressed genes, they are likely noise and have limited deleterious consequences. The authors should examine the changes of DEGs and DAS events relative to gene expression levels in most of their analyses.

More importantly, the authors need to identify the underlying mechanisms that lead to the DEGs and DAS events. Otherwise, these kinds of analyses have a very limited impact on future studies. For example, RNA editing of transcripts may lead to loss or gain of splicing sites, which may be the reason for some of the DAS events. The authors may also perform CLIP-seq to check whether Apobec and TadA tend to bind to the DEGs or the flanking region of DAS events.

To confirm the DEGs and DAS events are indeed due to Apobec or TadA expression, data from different cell lines should be collected and compared. If the authors' hypothesis is true, consistent changes should be observed in different cell lines.

Minor points:

More discussions are needed to clarify how this finding may provide insight into the improvement of the ABE and CBEs to minimize the potential deleterious effect of DEG and DAS events.

Re: Transcriptome-wide effects of cytosine and adenine deaminases on base-editor induced gene expression (Manuscript ID COMMSBIO-20-3010A)

RESPONSE TO REVIEWERS' COMMENTS:

Reviewer #1 (Remarks to the Author):

Previous studies showed that cytosine or adenine base editors (CBEs or ABEs) induced substantial off-target effects at both DNA and RNA levels, which raised serious concerns about the therapeutic applications of these genome editing tools. In this study, the authors examined the transcriptome-wide deaminase effects on gene expression and found that both CBEs and ABEs generated hundreds to thousands differentially expressed genes (DEGs) and differential alternative splicing (DAS) events. They further examined the previously reported high-fidelity base editors and found that these editors only showed limited effects on the reduction of DEGs and DAS. Overall, I think this study reveals a very important aspect of base editor-induced off-target effects. However, the authors missed an essential control group for almost all sets of experiments: how many DEGs and DAS events could be induced by comparing GFP v.s. GFP cells (for reference, see PMID: 31181567)? This control group must be included for all comparisons.

Response: We sincerely thank the reviewer for the encouraging comments and for this very meaningful suggestion. In most experiments, we used GFP-only groups as controls when we identified DEGs and DAS events. We have described the information of controls we used for each experiment in the Methods, Results, and Figure Legends more clearly. According to reviewer's suggestion, we performed analysis of DEGs and DAS events comparing GFP versus GFP cells as an essential control group for other sets of experiments. We randomly selected three repeats from six repeats of GFP cells as control groups and the other three repeats as treatment groups. The numbers of DEGs comparing between the two GFP groups were closed to 0 (Supplementary Fig. 1a, Fig. 6c), and the numbers of DAS events were mostly less than 100 (Supplementary Fig. 3a, Fig. 6f). So these DEGs or DAS events we identified were indeed induced by deaminase enzymes.

Reviewer #2 (Remarks to the Author):

In this study, the authors systematically analyzed the potential impact of ABEs and CBEs on gene expression and alternative splicing. They found that both editors could induce hundreds of DEGs and DAS events, which seems independent of cas9 expression. This is an interesting topic to look into; however, the current results are too preliminary to draw any conclusions.

Response: Thank you very much for your kind comments. We also believe that our findings are novel and interesting.

Major points:

(1) The fold changes of DEGs and DAS events are only informative when used in conjunction with their gene expression levels. For example, if most of these events occur in lowly expressed genes, they are likely noise and have limited deleterious consequences. The authors should examine the changes of DEGs and DAS events relative to gene expression levels in most of their analyses.

Response: We thank the reviewer for raising this question. We analyzed the expression levels of DEGs and DAS events by calculating the rates of expressed genes. Here we defined expression threshold the fragments per kilobase of transcript per million mapped reads (FPKM) >1 in at least one repeat of control or BE-treated groups to identify an expressed DEG or DAS event. We found that the rates of expressed genes of DEGs were about 50% to 70% (Supplementary Fig. 1f, 1g, 2c, 2f, 2g) and more than 90% of DAS events occurred to expressed genes (Supplementary Fig. 3b-f), which means that these DEGs and DAS events may have confounding effects or consequences.

(2) More importantly, the authors need to identify the underlying mechanisms that lead to the DEGs and DAS events. Otherwise, these kinds of analyses have a very limited impact on future studies. For example, RNA editing of transcripts may lead to loss or gain of splicing sites, which may be the reason for some of the DAS events. The authors may also perform CLIP-seq to check whether Apobec and TadA tend to bind to the DEGs or the flanking region of DAS events.

Response: We sincerely thank the review for pointing out this issue. Many studies have been exploring the potential mechanism for the generation of off-target effects induced by base editors. Here we have also carried out experiments to identify the underlying mechanisms that lead to the DEGs and DAS events on the advice of editor and reviewer. While CLIP is an antibody-based technique used to study RNA-protein interactions related to RNA immunoprecipitation, this is very hard for us to get specific and effective anti-APOBEC1 or anti-TadA antibodies in a short time.

RNA editing of transcripts may lead to loss or gain of splicing sites, which may be the reason for some of the DAS events. According to editor and reviewer's suggestion, we detected whether candidate deaminase editing events will map to a particular DEG or DAS gene. We have referred to literatures and papers such as Lo Giudice *et al.* (2020a)¹ suggested by editor. *De novo* RNA editing candidates were called from the RNA-seq data in each replicate separately, following the detection and filtering procedure described in Lo Giudice *et al.* (2020a)¹, Lo Giudice *et al.* (2020b)² and Gokul Ramaswam *et al.* (2013)³. The deep transcriptome data we used to identifying RNA editing sites was carried out using Illumina Hiseq at mean coverages of $125 \times$ ⁴. Briefly, raw data were initially inspected using FastQC (v0.11.8) and reads were trimmed with Trim Galore (v0.6.1) using default parameters for pair-end data to remove both low-quality reads and adaptors. Then, high quality reads were aligned onto the GRCh38 human transcriptome with STAR (v2.5.0a). After the preprocessing, RNA-seq reads were passed to the REDIttoolDenovo.py script from REDIttools package⁵ using parameters "-t 10 -e -d -l -U AG -p -u -m20 -T6-0 -W -v 1 -n 0.0" to identify RNA editing candidates. At the end of the run, all variants were further filters as follow: (i) Removed variant sites with variation frequency <0.1 ¹; (ii) Excluded variant sites with an extreme degree of variation (100%), which are likely genomic SNPs³; (iii) Removed variant sites identified in GFP-alone to find the base editor-induced editing sites; (iv) Removed variant sites identified as off-target SNVs from Zhou, C. *et al.* (2019)⁴.

Finally, we obtained 31, 35, 21 and 14 putative editing sites in cells among APOBEC1, BE3, BE3-site 3, and BE3-RNF2, respectively. The number of DEGs mapped to these consistent mismatches identified was 0, 4, 0 and 0 (the number of DEGs identified in each group was 1178, 757, 172 and 316 as shown in Fig. 1a). For DAS events, we extracted 4 bp flanking genomic sequences each side of differential

alternative splice sites. This sequence was then reverse-complemented and aligned against the editing sites. However, editing sites mapped to none of the DAS events identified in CBE groups (the number of DAS events identified in CBE groups was 1779, 568, 1821 and 1612 as shown in Fig. 3a). For ABEs, we detected 218, 82, 129 and 170 RNA editing sites in cells expressing only TadA-TadA*, ABE7.10, ABE7.10-site 1 and ABE7.10-site 2, respectively. The number of DEGs mapped to these consistent mismatches identified was 16, 4, 0 and 3 (the number of DEGs identified in ABE groups was 1761, 859, 215, and 357 as shown in Fig1.c). These consistent mismatches mapped to 11, 9, 6 and 7 of the DAS events identified in ABE groups (the number of DAS events identified in ABE groups was 1949, 1324, 513 and 1115 as shown in Fig. 3b).

In humans, the most prevalent type is adenosine-to-inosine (A-to-I) editing, catalyzed by the adenosine deaminase acting on RNA (ADAR) family of enzymes. We also detected the expression levels of ADAR1 and ADAR2 in base editor-treated groups as shown below. The expression of deaminases did not affect the levels of ADAR1 or ADAR2. Taken together, RNA editing of transcripts in base editor may not be the main mechanism leading to the effects in gene expression or splicing induced by deaminases.

Volcano plots showing the significant DEGs (adjusted p-value ≤ 0.05 and |Fold change| ≥ 2 ; downregulated (blue), upregulated (red), and unchanged genes (gray)) in

base editor-treated groups. The expression of ADAR1 and ADAR2 was labeled separately. Cells expressing GFP served as control.

We reasoned that higher expression of deaminases will induce more DEGs and DAS events. Thus, we examined the potential effect of the expression levels of deaminases by performing gradient concentration transfection of APOBEC1 or TadA-TadA* in HEK293T cells. The gradient expression of APOBEC1 or TadA-TadA* for each group was confirmed by qPCR and FPKM values calculated from RNA-seq data (Fig. 6a, b, d, e). We found that the numbers of DEGs and DAS events were increased when higher levels of APOBEC1 or TadA-TadA* were expressed. There were 0, 990, 2119 and 2581 DEGs identified in HEK293T cells transfected with 0 μ g, 1.5 μ g, 2.5 μ g or 3.5 μ g APOBEC1, respectively (Fig. 6c, Supplementary Data 14). Totals of 146, 652, 1061 and 1346 DAS events were observed in gradient concentration of APOBEC1-transfected groups, respectively (Fig. 6c, Supplementary Data 14). Simultaneously, we examined the potential effect of expression of ABE deaminases by performing gradient concentration transfection of TadA-TadA* in HEK293T cells (Supplementary Fig. 4a) and confirmed the expression levels of deaminases in each group (Fig. 6d, e). We also observed an increase number in DEGs and DAS events induced by increased expression of ABE deaminases (Fig. 6f, Supplementary Data 14). There were 0, 1264, 1076 and 2312 DEGs and 146, 956, 1384 and 1607 DAS events identified in HEK293T cells transfected with 0 μ g, 1.5 μ g, 2.5 μ g or 3.5 μ g TadA-TadA*, respectively (Fig. 6f). These data confirmed our supposition that higher expressing level of CBEs or ABEs will result in more DEGs and DAS events. Note that this hypothesis does not preclude the possibility that there are other potential mechanisms for these widespread changes induced by base editors. Future studies are necessary to explore the proposed mechanism and identify potential unanticipated activities of base editors.

References

1. Lo Giudice, C., Tangaro, M.A., Pesole, G. & Picardi, E. Investigating RNA editing in deep transcriptome datasets with REDIttools and REDIportal. *Nat Protoc* **15**, 1098-1131 (2020).
2. Lo Giudice, C. *et al.* Quantifying RNA Editing in Deep Transcriptome Datasets. *Front Genet* **11**, 194 (2020).
3. Ramaswami, G. *et al.* Identifying RNA editing sites using RNA sequencing data alone. *Nat Methods* **10**, 128-132 (2013).

4. Zhou, C. *et al.* Off-target RNA mutation induced by DNA base editing and its elimination by mutagenesis. *Nature* **571**, 275-278 (2019).
5. Picardi, E. & Pesole, G. REDIttools: high-throughput RNA editing detection made easy. *Bioinformatics* **29**, 1813-1814 (2013).

(3) *To confirm the DEGs and DAS events are indeed due to Apobec or TadaA expression, data from different cell lines should be collected and compared. If the authors' hypothesis is true, consistent changes should be observed in different cell lines.*

Response: We thank the reviewer for this valuable suggestion. It is very necessary to test the applicability of our findings to other mammalian cell types. We observed the changes in gene expression and splicing induced by base editors in HeLa cells and there were 1,386 and 732 DEGs identified in BE3 or TadA-TadA* transfected HeLa cells, including 34 and 31 cancer-related genes, respectively (Fig. 5a-b, Supplementary Fig. 6a, b, Supplementary Data 13). Totals of 2,035 and 1,263 DAS events were observed in BE3 and TadA-TadA*-treated groups, respectively (Fig. 5c, Supplementary Data 13). We also randomly selected several DEGs and DAS events and verified the differential gene expression and splicing in HeLa cells (Fig. 5d-i). Together, these data indicate that the transcriptome-wide DEG and DAS events induced by base editors ubiquitously occur in mammalian cells.

Minor points:

More discussions are needed to clarify how this finding may provide insight into the improvement of the ABE and CBEs to minimize the potential deleterious effect of DEG and DAS events.

Response: We thank the reviewer for this suggestion. We discussed in depth on the potential effect of DEGs and DAS events induced by base editors in Discussion section of our manuscript as follow:

Expect for off-target RNA SNVs effect induced by genome-editing technologies reported by recent studies, we show here a critical and previously unreported aspect of the effects deaminases impose on gene expression and alternative splicing. We present further insight into the current understanding of the potential effects of

deaminases induced by DNA base editors, which has important implications for the application of these technologies in both research and clinical settings. The most basic requirement of application the genome-editing technologies in clinical treatment is accuracy and safety. Safety assessments for human therapeutic applications should include a comprehensive evaluation of the potential functional consequences induced by transcriptome-wide DEGs and DAS events. Although these differentially expressed genes or differential alternative splicing events could exist for only a short period of time by transient expression of base editors, the longer-term functional consequences of widespread gene expression and alternative splicing, and the confounding effects of unwanted DEGs and DAS events will need to be accounted for in research studies. We have supplemented more discussions in the revised manuscript.

Reviewers' comments:

Reviewer #1 (Remarks to the Author):

I am satisfied with the revision and support the publication of this study.

Reviewer #2 (Remarks to the Author):

This revised manuscript is improved and generally more cohesive. Unfortunately, however, the newly provided results and analyses did little to mitigate my main concerns. Namely, I still find that the authors have not convincingly demonstrated the impact of ABEs and CBEs on gene expression and alternative splicing. Moreover, the newly added editing site analyses raise new concerns about the validity of the experiments.

Remaining concerns that were unaddressed or related to newly provided results and responses include:

1. "Authors' Response: While CLIP is an antibody-based technique used to study RNA-protein interactions related to RNA immunoprecipitation, this is very hard for us to get specific and effective anti-APOBEC1 or anti-TadA antibodies in a short time. " - As far as I know, anti-APOBEC1 antibodies are commercially available and work well for CLIP experiment.
2. In the newly added editing call analyses, the authors only identified <100 BE3 sites and <300 ABE sites. These observations are inconsistent with the original publications, which identified tens of thousands of BE3 or ABE induced editing sites. More analyses need to be done to support the validity of the experiments, and possible causes for this inconsistency need to be identified.
3. Figure 5: to confirm that the DEGs and DAS events are indeed due to Apobec or TadA expression, the authors need to show the overlaps of DEG or DAS events between 293 and HeLa cells.

RESPONSE TO REVIEWERS' COMMENTS:

1. “Authors’ Response: While CLIP is an antibody-based technique used to study RNA-protein interactions related to RNA immunoprecipitation, this is very hard for us to get specific and effective anti-APOBEC1 or anti-TadA antibodies in a short time.” - As far as I know, anti-APOBEC1 antibodies are commercially available and work well for CLIP experiment.

Response: We appreciate for your very meaningful advice and we also confirmed the antibodies are available for CLIP-seq. Upon receiving your suggestions, we contacted top companies in Next Generation Sequencing (NGS) market in China (Table R1), including BGI, Novogene, Annoroad and CapitalBio Technology Corporation. However, as CLIP-seq is a less commonly used technique, these sequencing companies can’t perform CLIP-seq. We also contacted several small-scale sequencing companies and two of them replied that they could try to perform CLIP-seq library preparation and sequencing, but they were unable to ensure the success of the experiment. At the same time, we also contacted laboratories in China that maybe capable of performing CLIP-seq, including Prof. Mingxi Liu from State Key Laboratory of Reproductive Medicine, Nanjing Medical University, they also could not guarantee the success of CLIP-seq experiment. We agree with your constructive suggestion about performing CLIP-seq to identify the direct target of deaminase, but it is extremely hard for us to conduct the experiment at this stage.

Table R1. The companies we contacted for CLIP-seq

Company	Email	About CLIP-seq
CapitalBio Technology Corporation, Beijing, China	zhaopin@capitalbiotech.com	They can’t perform CLIP-seq
BGI-Shenzhen, Beishan Industrial Zone, Shenzhen, China	info@genomics.cn	They can’t perform CLIP-seq
Novogene Co., Ltd, Building 301, Beijing, China	service@novogene.com	They can’t perform CLIP-seq
Annoroad Gene Technology Co.,	office-bj@annoroad.com	They can’t perform CLIP-seq

Ltd, Beijing, China		
BioGenius Bio technology Co., Ltd, Shanghai, China	service@biogenius.cn	They can try to perform CLIP-seq library preparation and sequencing, but can't ensure the success of experiment
Seqhealth Tech Co. Ltd, Wuhan, China	seqhealth@seqhealth.cn	They can try to perform CLIP-seq library preparation and sequencing, but can't ensure the success of experiment

Although we cannot perform CLIP-seq experiment as you required under current situation, we really hope you can believe that we have made sufficient work to draw our definitive findings of this study. We analyzed several independent datasets from different libraries, including Hui Yang and David Liu and these datasets included many different versions of base-editors. In addition, we performed RNA-seq in 293T and HeLa cells to verify the differential expression genes and splicing by PCR in our library. We also performed gradient concentration to transfect deaminases in our revised manuscript to confirm our hypothesis that higher expression levels of CBEs or ABEs result in more DEGs and DAS events. Taken together, we provide solid evidence to show that DEGs and DAS events are generated in the base-editor-treated cells including the engineering of deaminases-treated cells.

Currently, much attention has been focused on the genome-wide and transcriptome-wide off-target mutations in BEs, and great improvements have been made to eliminate the RNA off-target activity of BEs. However, these studies all ignored an important aspect that BEs will introduce transcriptome-wide DEGs and DAS events in addition to their observed off-target RNA SNVs. Similar to the poorly understanding for the off-target mutations introduced by BEs, we cannot present the exact and direct mechanism of the DEGs or DAS events introduced by BEs, which is

a weak point for our current study. However, we believe that our findings still are very instructive and informative for future studies.

2. In the newly added editing call analyses, the authors only identified <100 BE3 sites and <300 ABE sites. These observations are inconsistent with the original publications, which identified tens of thousands of BE3 or ABE induced editing sites. More analyses need to be done to support the validity of the experiments, and possible causes for this inconsistency need to be identified.

Response: We should first apologize for our misunderstanding of your comment. In our previous revision, we misunderstood RNA-editing events, as you indicated, to be the RNA editing sites catalyzed by the adenosine deaminase acting on RNA (ADAR) family of enzymes. Therefore, we identified RNA editing candidates using parameters "-t 10 -e -d -l -U AG -p -u -m20 -T6-0 -W -v 1 -n 0.0" and **filtered out variant sites including both public dbSNP database and off-target SNVs**. That is to say, our identified RNA editing sites were not included the RNA off-target SNVs, i.e., RNA-editing events that you mentioned. Now we re-analyzed editing sites according to your advice. The detection and filtering procedure are shown as below:

Raw data were initially inspected using FastQC (v0.11.8) and reads were trimmed with Trim Galore (v0.6.1) using default parameters for pair-end data to remove both low-quality reads and adaptors. Then, high quality reads were aligned onto the GRCh38 human transcriptome with STAR (v2.5.0a). After the preprocessing, RNA-seq reads were passed to the REDIttoolDenovo.py script from REDIttools package using parameters "-t 10 -e -d -l -p -u -m20 -T6-0 -W -v 1 -n 0.0" to identify RNA editing candidates. At the end of the run, all variants were further filters to removed variant sites identified in GFP-alone to find the base editor-induced editing sites.

Finally, we obtained 17826, 12393, 6530 and 8893 putative editing sites in cells among APOBEC1, BE3, BE3-site 3, and BE3-RNF2, respectively. The number of DEGs mapped to these consistent mismatches identified was 60, 26, 9 and 15 (the number of DEGs identified in each group was 1178, 757, 172 and 316 as shown in

Fig. 1a). For DAS events, we extracted 4 bp flanking genomic sequences each side of differential alternative splice sites. This sequence was then reverse-complemented and aligned against the editing sites. These editing sites mapped to 81, 9, 30 and 23 of the DAS events identified in APOBEC1, BE3, BE3-site 3, and BE3-RNF2, respectively (the number of DAS events identified in CBE groups was 1779, 568, 1821 and 1612 as shown in Fig. 3a).

Table R2. The RNA editing sites identified in CBEs

CBE groups	APOBEC1	BE3	BE3-site 3	BE3-RNF2
Editing sites	17826	12393	6530	8893
DEGs	1178	757	172	316
DEGs mapped to editing sites	60 (5.1%)	26 (3.4%)	9 (5.2%)	15 (4.7%)
DAS events	1779	568	1821	1612
DAS mapped to editing sites	81 (4.6%)	9 (1.6%)	30 (1.6%)	23 (1.4%)

For ABEs, we detected 7501, 3495, 3505 and 6661 RNA editing sites in cells expressing only TadA-TadA*, ABE7.10, ABE7.10-site 1 and ABE7.10-site 2, respectively. The number of DEGs mapped to these consistent mismatches identified was 106, 44, 19 and 28 (the number of DEGs identified in ABE groups was 1761, 859, 215, and 357 as shown in Fig. 1c). These consistent mismatches mapped to 86, 33, 11 and 44 of the DAS events identified in TadA-TadA*, ABE7.10, ABE7.10-site 1 and ABE7.10-site 2, respectively (the number of DAS events identified in ABE groups was 1949, 1324, 513 and 1115 as shown in Fig. 3b).

Table R3. The RNA editing sites identified in ABEs

ABE groups	TadA-TadA*	ABE7.10	ABE7.10-site 1	ABE7.10-site 2
Editing sites	7501	3495	3506	6661
DEGs	1761	859	215	357
DEGs mapped to editing sites	196 (11.1%)	44 (5.1%)	19 (8.8%)	28 (7.8%)
DAS events	1949	1324	513	1115
DAS mapped to editing sites	86 (4.4%)	33 (2.5%)	11 (2.1%)	44 (3.9%)

3. Figure 5: to confirm that the DEGs and DAS events are indeed due to Apobec or TadA expression, the authors need to show the overlaps of DEG or DAS events between 293 and HeLa cells.

Response: Thank you very much for your constructive comment. The overlaps of DEG and DAS events between 293T and HeLa cells are shown in Figure R1. The number of overlaps of DEG and DAS events between 293T and HeLa cells were well consistent with the observations of off-targets SNVs between different cell types. These results suggest that DEG and DAS events induced by base-editor tend to be essentially random as like RNA off-target SNVs.

Figure R1. Summary of DEG and DAS events between 293T and HeLa cells. a) Venn diagram showing numbers of DEGs identified in BE3-transfected HEK293T cells and HeLa cells. b) Venn diagram showing numbers of DEGs identified in TadA-TadA*-transfected HEK293T cells and HeLa cells. c) Venn diagram showing numbers of DAS events identified in BE3-transfected HEK293T cells and HeLa cells. d) Venn diagram showing numbers of DAS events identified in TadA-TadA*-transfected HEK293T cells and HeLa cells.

Reference

1. König J, Zarnack K, Luscombe NM, Ule J. Protein-RNA interactions: new genomic technologies and perspectives. *Nat. Rev. Genet.* 2011;13:77–83
2. Zhou, C. et al. Off-target RNA mutation induced by DNA base editing and its elimination by mutagenesis. *Nature* 571, 275-278 (2019).

REVIEWERS' COMMENTS:

Reviewer #2 (Remarks to the Author):

I am now satisfied with the revision and support the publication of this study.